# Toward Explainable Physical Audiovisual Commonsense Reasoning

## ABSTRACT

For AI systems to be safely and reliably grounded in the real world, they should possess the ability of physical commonsense reasoning, *i.e.* they are desired to understand the physical properties, affordances, and maneuverability of objects in everyday life. Physical commonsense reasoning is essentially a multisensory task as physical properties of objects are manifested through multiple perception modalities, including both visual and auditory. In this study, we constructed two new benchmarks, called PACS-Reason and PACS-Reason+, for explainable physical audiovisual commonsense reasoning (EPACS), in which each datapoint is accompanied by a golden detailed *rationale* (*intermediate reasoning path*) to explain the answer selection. Moreover, we present PAVC-Reasoner, a multimodal large language model (LLM) designed to reason about physical commonsense attributes. The model aligns different modalities with the language modality by integrating three different *perceivers* for cross-modal pretraining and instruction finetuning at multiple granularities. It utilizes an LLM as a cognitive engine to process multimodal inputs and output convincing intermediate reasoning paths as *justification* for inferring answers. Numerous experiments have demonstrated the effectiveness and superiority of PAVC-Reasoner as a baseline model for studying EPACS. Most attractively, PAVC-Reasoner is capable of reasoning and obtaining strong interpretable explicit reasoning paths, signifying a significant stride towards real-world physical commonsense reasoning.

## CCS CONCEPTS

• **Computing methodologies → Knowledge representation and reasoning**.

## KEYWORDS

multimodal commonsense reasoning, commonsense reasoning, explainable reasoning, physical audiovisual commonsense reasoning

## 1 INTRODUCTION

Physical commonsense represents a ubiquitous comprehension of the physical properties and affordances of everyday objects [5]. Humans effortlessly acquire physical commonsense and engage in commonsense reasoning through daily life experiences and observations, whether it is to infer properties of objects (*e.g.* "*one might tap the rind of a watermelon to assess the sound because people know*

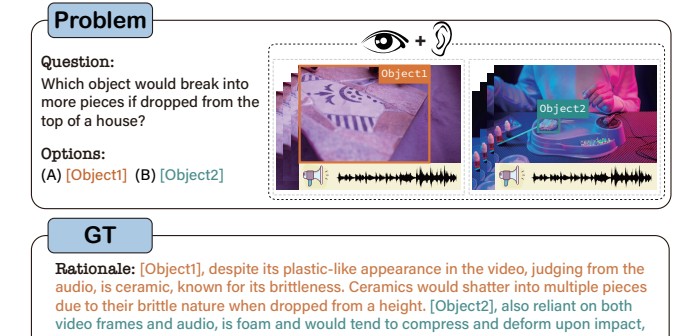

**Figure 1: Illustration of two datapoints from PACS-Reason. Each datapoint contains a pair of objects and a question. Here Object1 is made of ceramic and Object2 is made of foam.**

*that ripe watermelons make a resonant sound while unripe watermelons make a dull thumping sound.*") or solve unique problems (*e.g.* "*in the absence of using a hammer to break the window, should I use a suitcase or a pillow?*") [51]. How can artificial intelligence (AI) systems reason about the physical world without experiencing it firsthand? The answer to this question may well be the key to moving towards general artificial intelligence (GAI). Undoubtedly, to ensure the safe and reliable deployment of AI systems in real-world scenarios such as robot navigation and autonomous driving, they must understand the *physical properties*, *affordances of everyday objects*, *interactions with other objects*, and *how to manipulate these objects* [5, 16, 51].

Physical commonsense reasoning is, by its very nature, a multisensory task as physical properties can be expressed through multiple modalities, including vision and acoustic [21, 23, 40, 42, 46]. If two objects appear visually similar, audio can provide valuable information for distinguishing the physical characteristics between them. As illustrated in Fig. 1, based on visual observation alone, Object1 may be mistaken for *plastic* or even *paper*, while Object2 may be misidentified as *wax*. That is, the lack of necessary audio information may lead to an inaccurate answer to question (Q1), which involves the query about the fragility of Object2 relative to Object1 when dropped from a certain height. Thus, to endow machines with physical commonsense reasoning, they must first possess the ability to perceive and understand multimodal signals.

Human cognition, when addressing commonsense questions, harnesses information from both the auditory and visual modalities to construct coherent and comprehensive chains of thought. For instance, when responding to Fig. 1 Q1, individuals initially need to discern the composition of Object1 and Object2 from the audiovisual content provided by the video pair. Subsequently, based on the intrinsic physical properties of the objects, one can infer that Object1 (ceramic) would fragment into more pieces when

 

dropped from the top of a house compared to Object2 (foam). *How can we enable machines to replicate, or at least manifest, such explicit reasoning processes akin to human cognition?* A straightforward idea is to articulate the cognitive thought chains of the human brain in natural language and then use them to teach machines. Adhering to this idea, we have established two novel benchmarks, PACS-Reason and PACS-Reason+, building upon the original PACS [51], which stands as the pioneer in audiovisual benchmarks for the annotation of physical commonsense attributes. Each datapoint in PACS [51] is a value tuple $(q, o_1, o_2, l)$, representing the question, two objects, and a binary label indicating which object is the correct answer. We augment each data point in PACS by appending a detailed *rationale* (natural language explanation) for each question-answer pair. In PACS-Reason, we offer these rationales (*i.e.* intermediate reasoning paths) as new ground truth, as illustrated in Fig. 1. These rationales undergo careful human scrutiny and serve as explicit reasoning paths, dramatically boosting the model's understanding of the physical commonsense attributes in each datapoint.

To sum up, the main contributions of this work are as follows: (1) We establish two enhanced benchmarks, namely PACS-Reason and PACS-Reason+, for *explainable physical commonsense reasoning* (EPACS). These benchmarks offer meticulously crafted detailed explanations for answer choices, allowing for a comprehensive evaluation of the physical commonsense reasoning capabilities of multimodal models. This signifies a significant stride in reasoning about the physical world. (2) We introduce PAVC-Reasoner, a multimodal foundation large model for audio-visual commonsense reasoning. This model simultaneously considers modality synergy and specificity, leveraging a large language model (LLM) as the core for processing multimodal signals and generating *natural language rationales*. Specifically, the model integrates three *perceivers* to bridge multimodal inputs and LLM and achieves alignment between different modalities and the language modality through multi-grained cross-modal pretraining and instruction fine-tuning. (3) Extensive experiments demonstrate the effectiveness and superiority of PAVC-Reasoner. It demonstrates a precise understanding of the material composition, physical attributes, and availability of objects in videos, and achieves state-of-the-art performance on the two newly constructed EPACS benchmarks.

## 2 RELATED WORK

### 2.1 Enhanced Reasoning with Explanations

An increasing number of vision-language reasoning tasks demand models not only capable of inferring answers to questions but also possessing robust explanatory capabilities [7, 9, 26, 30, 32, 45]. Among them, Explanatory Visual Question Answering (EVQA) [9] emerges as a burgeoning multimodal reasoning task, aiming to require models to generate multimodal explanations about the answers while answering visual questions. In contrast to traditional VQA tasks, EVQA emphasizes offering user-friendly explanations to enhance the interpretability and reliability of reasoning models. Methods for enhancing vision-language reasoning with explanations be summarized as follows: 1) *Symbolic rule-based explanatory model* [7, 49]: Yi et al. introduced a model that entirely decouples vision-language understanding from reasoning. Here, deep representation learning is employed for vision-language understanding,

while executable symbolic programs are utilized for reasoning. 2) *Natural language explanatory model* [9, 32]: Explanations in natural language form offer high-level, intuitive interpretations that are easily understandable to humans, supplementing low-level interpretations based on visual perception and semantics. One of the representative works is the Rationale Transformer [32], which integrates pretrained language models with object recognition, grounded visual semantic frames, and visual commonsense graphs to learn to generate explanations in free-text form. 3) *Causal explanation-augmented models*[30, 33, 45]: Methods along this line aim to identify true causal relationships between question-answer pairs or answer-explanations, or among object features, through causal models. For example, Lv et al. introduced a counterfactual learning module to enhance the model's commonsense reasoning abilities by simulating physical knowledge relationships between different objects under counterfactual interventions.

### 2.2 Multimodal Large Language Model

Recent advances in Large Language Models (LLMs), exemplified by ChatGPT [34] and GPT-4 [1], have demonstrated remarkable language understanding and reasoning capabilities. The flourishing development of LLMs has spurred advancements in multimodal learning and given rise to the emergence of Modal Large Language Models (MLLMs) [12, 25, 28, 29, 31, 47, 55, 60]. These models typically adhere to the following paradigm: initially transforming multimodal inputs into text or mapping them into text embedding spaces, then leveraging LLMs as the brain/controller for processing multimodal information and providing reasoning results. MLLMs inherit three key capabilities of LLMs - In-Context Learning (ICL) [6], Instruction Following [35], and Chain-of-Thought (CoT) [44], demonstrating outstanding performance in intricate multimodal tasks. For instance, Flamingo [2] enables multimodal ICL, adapting to new tasks with minimal demonstrations. LLaVA [29] achieves GPT-4-like instruction following capabilities through vision-language instruction fine-tuning, while PaLM-E [14] demonstrates a range of multimodal capabilities related to robotics, including perception and planning, through zero-shot multimodal chain-of-thought (CoT) reasoning. These emerging capabilities of MLLMs shed light on our journey towards interpretable physical world reasoning.

## 3 RATIONALE GENERATION WITH PAVC-REASONER

**Preliminary** Recent works [27, 50, 56] confirm the rich commonsense and world knowledge embedded within LLMs. This study aims to leverage the commonsense knowledge and text-generation capabilities of LLMs to provide detailed explanations for physical audiovisual commonsense reasoning (PACS) tasks. Considering the multimodal nature of the PACS task, we employ outputs from three distinct perceiver branches to enhance the input of LLMs, thereby improving their multimodal perception and understanding abilities.

### 3.1 Framework of PAVC-Reasoner

Fig. 2 shows that the perception module of PAVC-Reasoner primarily consists of three perceivers, *i.e.*, the *vision-language perceiver*, the *audio-language perceiver*, and the *audiovisual-language perceiver*. Below, we delve into the construction details of each perceiver.

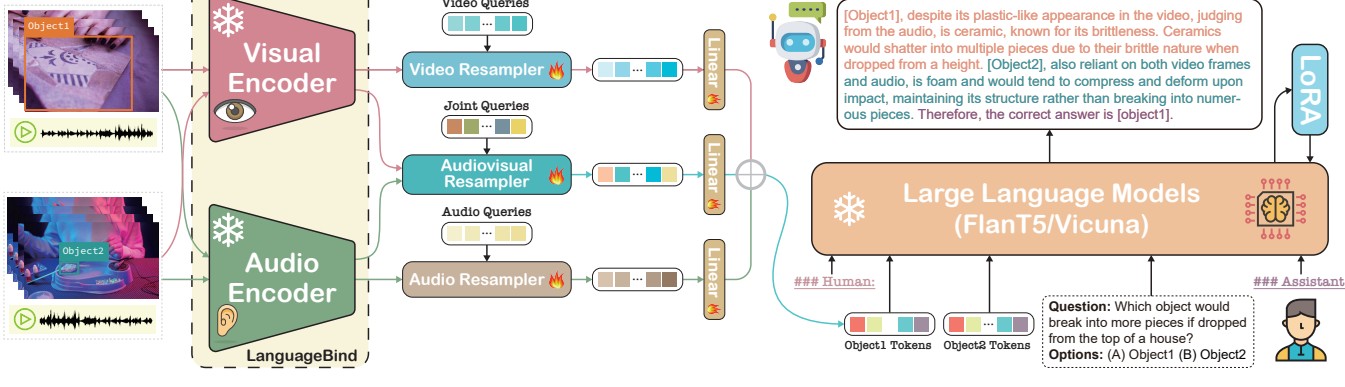

**Figure 2: Overall architecture of PAVC-Reasoner, which comprises three perceivers and a frozen LLM. Specifically, the *visual-language (V-L) perceiver* consists of a frozen visual encoder, a trainable `Video Resampler`, and a linear projection layer. This branch is responsible for extracting a fixed number of visual token representations and mapping them to the text embedding space of the LLM. The structure of the *audio-language perceiver* mirrors that of the V-L perceiver. The *audiovisual-language perceiver* contains the two frozen unimodal encoders, a trainable `Audiovisual Resampler`, and a linear projection layer, aiming to extract a fixed number of joint audiovisual representations and map them to the text embedding space of the LLM.**

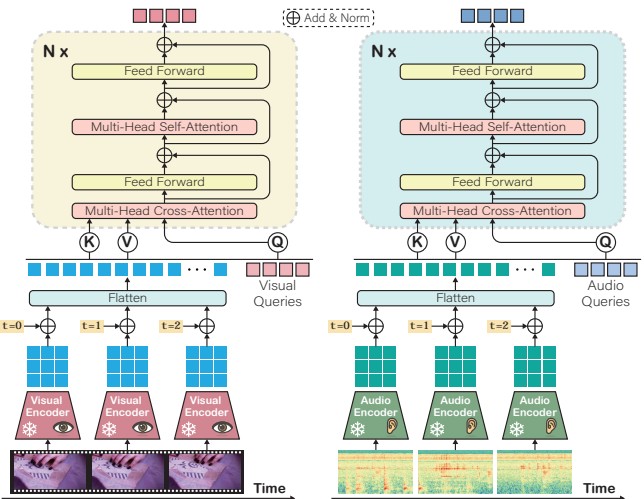

**Figure 3: Architecture of `Video/Audio Perceiver`.**

#### 3.1.1 Vision-Language Perceiver.

The aim of the *vision-language perceiver* is to endow LLM with visual perceptual capabilities. Specifically, for a given audible video, we sample T frames from it. The image encoder maps each frame $i$ to a frame feature $v_i \in \mathbb{R}^{n_v \times d_v}$, yielding a set of video frame features $\boldsymbol{v} = [v_1, v_2, \ldots, v_T]$. Similar to ViT [13], we add learnable position embeddings to the frame features. Subsequently, a flattening operation is performed on the set of video frame features $\boldsymbol{v} \in \mathbb{R}^{T \times n_v \times d_v}$ to obtain the video representation $\mathbf{V} \in \mathbb{R}^{(T \times n_v) \times d_v}$. Next, we input the position-encoded video representation $\hat{\boldsymbol{v}}$ into the `Video Resampler`. Fig. 3 illustrates the architectural details of the `Video Resampler`. Similar to the role of Q-Former in BLIP [24], `Video Resampler` uses $q_v$ video queries to obtain the video query embeddings $\mathbf{Q}_v \in \mathbb{R}^{q_v \times d_v}$ of dimension $d_v$. To align the dimension of the video representations with the dimension of the LLM text input (denoted as $d_t$), we introduce a

linear projection layer to convert the video query embedding $\mathbf{Q}_v$ into the *video prompt embedding* $\tilde{\mathbf{Q}}_v \in \mathbb{R}^{q_v \times d_t}$.

#### 3.1.2 Audio-Language Perceiver.

The *audio-language perceiver* aims to endow LLM with the capability to process and comprehend audio inputs. Given a piece of raw audio, it is first partitioned into M audio segments of $t$ seconds. For each segment, the audio spectrogram is obtained by computing a 128-dimensional log Mel filterbank, using a 25 ms Hamming window sliding every 10 ms intervals, resulting in a $128 \times 100t$-dimensional spectrogram. Then, the spectrogram is sliced into $n_a$ $16 \times 16$ patches, with temporal and frequency dimension strides both set to 10. Here, $n_a = 12\lceil(100t - 16)/10\rceil$ represents the number of patches and the effective input sequence length for the Transformer. Subsequently, each audio spectrogram $m$ is fed into a frozen audio encoder to generate an audio segment feature $a_m \in \mathbb{R}^{n_a \times d_a}$, thus creating a set of audio features $\boldsymbol{a} = [a_1, a_2, \ldots, a_M]$. Similar to the processing of `video resampler`, learnable positional embeddings [13] are added to audio segment features to enhance their temporal coherence. After flattening the set of the audio segment features $\boldsymbol{a} \in \mathbb{R}^{M \times n_a \times d_a}$, an audio representation $\mathbf{A} \in \mathbb{R}^{(M \times n_a) \times d_a}$ is obtained. The remaining processing flow is similar to that of the *visual-language perceiver*.

#### 3.1.3 Audiovisual-Language Perceiver.

Recent research in audiovisual scene understanding has underscored the importance of learning joint audiovisual representations for enhancing comprehension of free-form video content[18, 57, 58]. The goal of the *audiovisual-language perceiver* is to endow LLM with the capability to understand joint audiovisual content. As shown in Fig. 3, the video and audio representations $\mathbf{V}$ and $\mathbf{A}$ processed by the unimodal encoders and positional encoding layers are concatenated and fed into `Audiovisual Resampler`, which employs $q_k$ joint queries to learn the audiovisual query embedding $\mathbf{Q}_k \in \mathbb{R}^{q_k \times d_k}$ of dimension $d_k$. A linear layer is then used to map the audiovisual query embedding into the *audiovisual prompt embedding* $\tilde{\mathbf{Q}}_k \in \mathbb{R}^{q_k \times d_t}$ to keep the same dimension as the text embedding of the LLM.

## 3.2 Cross-modal Pretraining and Instruction Finetuning

We train our PAVC-Reasoner in two stages. **Stage I**, `cross-modal pretraining`, aims to bridge the gap between the multimodal output space and the embedding space of the LLM, achieving alignment between vision/audio and language to endow the large language model with foundational multimodal perception and comprehension capabilities. **Stage II**, `instruction finetuning`, seeks to enhance the model's instruction-following abilities, providing appropriate responses based on different instructions, as well as improving its reasoning and explanatory abilities in tasks involving understanding of audiovisual physical commonsense attributes.

### 3.2.1 Stage I: Cross-modal Pretraining.

The pretraining objective in this stage entails autoregressive language modeling, wherein the model learns to generate subsequent predictions of textual tokens based on multimodal contexts, maximizing the log-likelihood of generating textual tokens. In this stage, we consider $\mathbf{x}_v$ and $\mathbf{x}_a$ as a video clip and an audio clip input into the PAVC-Reasoner, respectively. Let $\mathbf{x}_q$ and $\mathbf{x}_r$ denote the questions (instructions) and ground-truth answers (responses) associated with audible videos, respectively, which can be represented by sequences of discrete textual tokens. We conduct cross-modal pretraining separately for three *perceivers*. Specifically, for the *vision-language perceiver*, the probability of generating $\mathbf{x}_r$ of a sequence of length L can be computed as follows:

$$p(\mathbf{x}_r|\mathbf{x}_v, \mathbf{x}_q) = \prod_{i=1}^{L} p_{\theta}(x_i|\mathbf{x}_v, \mathbf{x}_q, \mathbf{x}_{r,<i}), \quad (1)$$

where $\theta$ denotes trainable parameters, and $\mathbf{x}_{r,<i}$ represents the preceding answer tokens before the current prediction token $\mathbf{x}_i$. Two types of pretraining tasks are performed: *video-description generation* and *image-caption generation*. We treat images as single-frame videos. We leverage the Webvid-2M dataset [4], a large-scale collection of video clips sourced from the web, annotated with captions. In addition, we apply the LLaVA-CC3M-Pretrain-595K [29], a subset of the CC3M [39] image-text dataset. Compared to its original dataset, LLaVA-CC3M-Pretrain-595K has been curated to ensure a more balanced coverage of visual concepts. During pretraining, only the parameters of the `Video Resampler`, position embedding layer, and linear projection layer are trainable.

To pretrain the *audio-language perceiver* and generate an answer $\mathbf{x}_r$ with textual tokens of length L, the probability is as follows:

$$p(\mathbf{x}_r|\mathbf{x}_a, \mathbf{x}_q) = \prod_{i=1}^{L} p_{\psi}(x_i|\mathbf{x}_a, \mathbf{x}_q, \mathbf{x}_{r,<i}), \quad (2)$$

where $\psi$ denotes the trainable parameters of the `Audio Resampler`, position embedding layer, and linear projection layer. Note that we employ LanguageBind [59] as the frozen audio encoder, which learns a unified embedding space for all modalities by regarding the language as the bind across different modalities. This encoder is trained on a large-scale audio-text dataset VIDAL-10M [59], achieving alignment between audio and text modalities. This suggests the inherent capability of our *audio-language perceiver* in interpreting audio signals. In this stage, we aim to enhance the ability of LLM to understand audio content details by maximizing Eq. (2). To this

end, we leverage several existing environmental sound datasets to construct an *audio-language instruction tuning* dataset.

Specifically, we consider the AVE [43], Audio Set [17], VGG-Sound [8], and ESC-50 [36] datasets. Given an audio clip and its associated audio event class ($\mathbf{x}_a, \mathbf{x}_\ell$), our objective is to generate a set of instructions $\mathbf{X}_q$ to prompt the LLM to accurately articulate the content conveyed by the audio. We curate a list of instructions by GPT-4 [1] and randomly select an instruction $\mathbf{x}_q$, such as "Briefly describe the audio." With the following prompts, we convert audio-label pairs into instruction-tuning versions: `### Human:` $\mathbf{x}_a$ `<EOC>` `Question:` $\mathbf{x}_q$ `<EOC> ### Assistant: This audio describes` $\mathbf{x}_\ell$ `<EOC>`. Here, $\mathbf{x}_\ell$ denotes the description of the audio event class, *e.g.*, "*door knocking*" or "*glass breaking*". `<EOC>` serves as a special token indicating the end of a chunk.

To pretrain the *audiovisual-language perceiver*, we maximize the generation probability of an answer sequence $\mathbf{x}_r$ as follows:

$$p(\mathbf{x}_r|\mathbf{x}_v, \mathbf{x}_a, \mathbf{x}_q) = \prod_{i=1}^{L} p_{\phi}(x_i|\mathbf{x}_v, \mathbf{x}_a, \mathbf{x}_q, \mathbf{x}_{r,<i}), \quad (3)$$

where $\phi$ are learnable parameters of the `Audiovisual Resampler`, position embedding layer and linear projection layer. We pretrained this *perceiver* using the auxiliary task PACS-Material from the PACS benchmark. The PACS-Material task focuses on comparing the material composition of objects. Given a pair of objects described in two audible videos, the task is to determine which object is more likely to be made of a specific material, e.g. "*Which object is more likely to be made of glass?*" To facilitate autoregressive training, we annotated each data point in the PACS-Material training set in the form $(q, o1, o2, \ell)$ and converted it into a version in instruction format $(q, o1, o2, t_\ell)$. Here, $t_\ell$ represents the textual description of the binary label $\ell$, such as "*The correct answer is Object1.*"

### 3.2.2 Stage II: Instruction Finetuning.

After the pretraining stage, we integrate the trainable LoRA [22] modules into the LLM and optimize the weights of the three perceivers and the LoRA modules in an end-to-end manner on the PACS-Reason dataset collected in Sec. 3.3. In this stage, the training objective remains autoregressive (AR) language modeling. In order to train the PAVC-Reasoner to both output the answer to a question and give a *rationale* for deriving the answer, we suffixed the questions in the PACS-Reason dataset with some instructions that can trigger an explanation. These questions, once posed, ask PAVC-Reasoner to generate detailed intermediate reasoning steps (*i.e.*, rationales) about the answer in a specific format. For example, a question such as "*Which object would hurt more to step on?* **Why?**" requires the model to infer the answer and generate a *rationale* based on the audible video inputs and instructions.

## 3.3 Establishing New Benchmarks for PACS

To advance research in multimodal commonsense reasoning, we leveraged the original data points from PACS and employed foundational visual models and templates (see Sup. A) to prompt GPT-4 to generate rationales, thereby constructing a new explainable physical audiovisual commonsense benchmark, namely PACS-Reason and PACS-Reason+.

We begin by introducing PACS, a video-based multimodal commonsense benchmark designed specifically to evaluate the ability

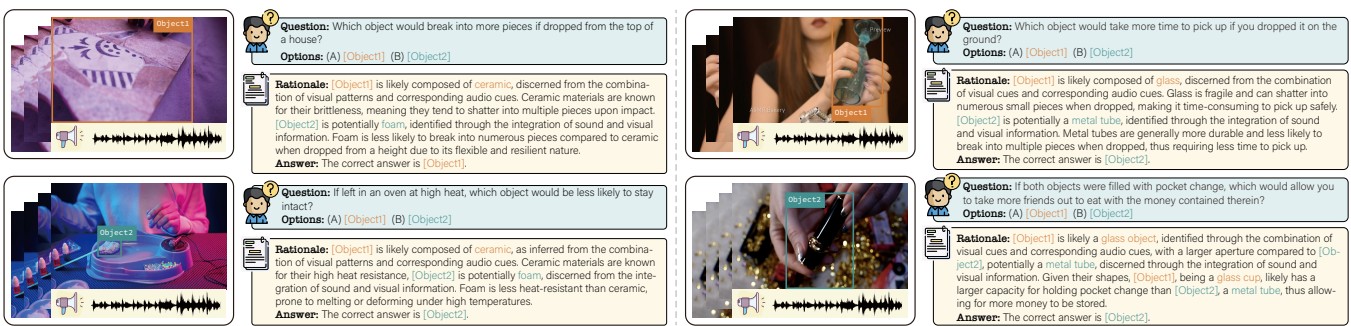

**Figure 4: Illustration of four datapoints from PACS-Reason, a new benchmark for interpretable physical audiovisual commonsense reasoning: Each datapoint consists of a pair of objects $o_1$, $o_2$, a question $q$, a binary label $\ell$ indicating which object is the correct answer, and a natural language rationale $r$ for explaining the answer selection.**

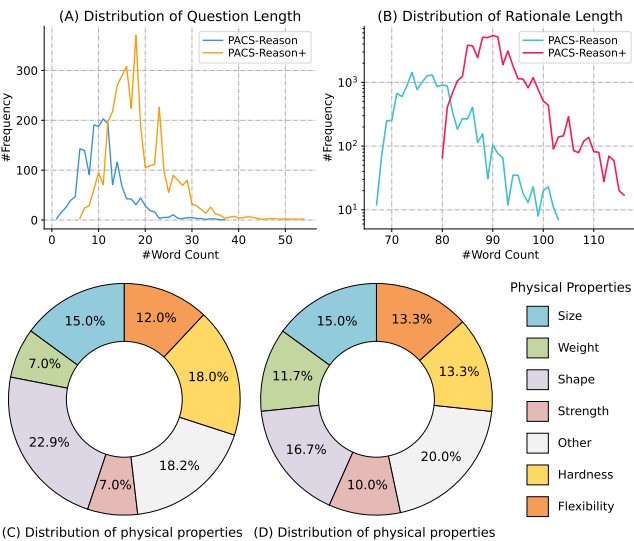

**Figure 5: Statistics of PACS-Reason and PACS-Reason+.**

**Table 1: Statistics of the datasets used in our experiments.**

| Benchmark | Partition | Datapoints | Videos | Rationale |
|---|---|---|---|---|
| PACS-QA[51] | train | 11,044 | 1,224 | |
| | val | 1,192 | 150 | ✗ |
| | test | 1,164 | 152 | |
| PACS-Material[51] | train | 3,460 | 1,224 | |
| | val | 444 | 150 | ✗ |
| | test | 445 | 152 | |
| **PACS-Reason** 🤖 | train | 11,044 | 1,224 | |
| | val | 1,192 | 150 | ✓ |
| | test | 1,164 | 152 | |
| **PACS-Reason+** 🤖 | train | 44,176 | 1,224 | |
| | val | 2,384 | 150 | ✓ |
| | test | 2,328 | 152 | |

of AI systems to reason about physical commonsense using audio and visual modalities. Its proxy task is binary question answering, where given a question $q$ and objects $o1$ and $o2$, the model must select the more appropriate object to answer the question. Each object is represented by a video $v$ showing interactions with the object, the corresponding audio $a$, and a bounding box $b$ drawn around the object in the intermediate frames of $v$. Hence, each data point in PACS is a tuple $(q, (b1, v1, a1), (b2, v2, a2), \ell)$, representing the question, two objects, and a binary label indicating which object is the correct answer.

**PACS-Reason** 🤖: It extends PACS-QA by augmenting each datapoint in its task with an additional thought chain, i.e., *detailed intermediate reasoning steps, serving as golden explanations for answer selection*. Specifically, each data point in PACS-QA is expanded into a new five-tuple $(q, o1, o2, \ell, r)$, where $r$ represents the rationale derived by the model for the answer. Hence, the proxy task for PACS-Reason can be defined as follows: given a pair of objects and a question, ***the model must select the more appropriate object to answer the question and offer a step-by-step thought chain for deducing the answer***, as illustrated in Fig. 4.

**PACS-Reason+** 🤖: Following the protocol of the original PACS, this benchmark introduces entirely new question-rationale pairs. Leveraging some foundational visual tools and GPT-4, this benchmark expands the original samples in the PACS dataset. To construct new physical commonsense questions, for a given pair of objects described in videos, we provide GPT-4 with detailed descriptions of two videos ($video1, video2$), the material compositions of object1 and object2, and the physical attributes associated with these objects. We prompt GPT-4 to generate new questions, answers, and their corresponding rationales based on these provided textual contexts and designed principles for new questions (see Sup. B), thus extending the original PACS benchmark in terms of physical attribute coverage and interoperability.

In Fig. 5, we illustrate the distribution of question lengths (in terms of word count) in PACS-Reason and PACS-Reason+. The questions in PACS-Reason have an average of 16.6 words, while those in PACS-Reason+ contain 18.4 words on average. Fig. 5 depicts the distribution of physical attributes involved in our questions. Compared to PACS-Reason, PACS-Reason+ demonstrates a more balanced distribution of physical attributes, largely mitigating the impact of data bias on models. Additionally, questions in PACS-Reason+ are more complex for the same pair of objects compared to PACS-Reason, desiring more intricate intermediate reasoning steps to derive the answers. Table 1 presents the statistical information of the PACS benchmarks used in our experiments.

**Table 2: Parameter counts of PAVC-Reasoner variants.**

| Models | Frozen Parameters ❄ | | | Trainable Parameters 🔥 | | | Count |
|---|---|---|---|---|---|---|---|
| | Language Model 🗣 | Vision Encoder 🖼 | Audio Encoder 🎙 | Video-Resampler | Audio-Resampler | Audiovisual-Resampler | |
| V1 | FlanT5$_{XL}$ (3B) | ViT-L/14 (303M) | AST (303M) | 103M | 103M | 194M | **4.1B** |
| V2 | FlanT5$_{XL}$ (3B) | ViT-G/14 (1.0B) | AST (303M) | 107M | 107M | 194M | **4.8B** |
| V3 | Vicuna (7B) | ViT-L/14 (303M) | AST (303M) | 103M | 103M | 194M | **8.1B** |
| V4 | Vicuna (7B) | ViT-G/14 (1.0B) | AST (303M) | 107M | 107M | 194M | **8.8B** |

**Table 3:** *Performance comparisons among diverse multimodal baselines on the PACS-Material/QA benchmarks.* **Results are reported as the mean and standard deviation of five runs.**

| Baseline Model | Subset | Accuracy (%) | | |
|---|---|---|---|---|
| | | PACS-Material | PACS-QA | Δ |
| *Simple Late Fusion Baseline* | | | | |
| Q+I+A+V [51] | val | 67.8±0.8 | 55.5±0.3 | 12.3 |
| | test | 67.4±1.5 | 55.0±1.1 | 12.4 |
| *Multimodal Foundation Large Models* | | | | |
| AudioCLIP [20] | val | 81.9±1.2 | 61.6±0.9 | 18.8 |
| | test | 75.9±1.1 | 60.0±0.9 | 15.0 |
| Merlot Reserve (B) [52] | val | 90.1±1.0 | 67.2±0.8 | 22.9 |
| | test | 85.2±1.4 | 66.5±1.4 | 18.7 |
| Merlot Reserve (L) [52] | val | 90.6±0.9 | 70.8±0.9 | 19.8 |
| | test | 87.3±1.5 | 70.1±1.0 | 17.2 |
| *Multimodal Large Language Models* | | | | |
| Video-LLaMA [54] | val | 76.3±1.0 | 63.6±1.2 | 12.7 |
| | test | 74.6±0.9 | 63.4±1.8 | 11.2 |
| VideoChat [25] | val | 80.1±0.8 | 62.9±1.4 | 17.2 |
| | test | 76.6±1.3 | 62.8±1.5 | 13.8 |
| Video-ChatGPT [31] | val | 83.1±1.1 | 64.4±1.2 | 18.7 |
| | test | 77.8±0.8 | 64.5±1.5 | 13.3 |
| Video-LLaVA [28] | val | 85.6±1.2 | 67.1±1.1 | 18.5 |
| | test | 81.0±0.9 | 66.6±1.2 | 14.2 |
| PAVC-Reasoner-V2 | val | 93.3±0.6 | 76.8±1.4 | 16.5 |
| | test | 88.0±0.8 | 76.2±1.5 | 11.8 |
| PAVC-Reasoner-V4 | val | 94.6±0.5 | 79.0±1.2 | 15.6 |
| | test | 91.6±0.8 | 78.9±0.9 | 12.7 |

## 4 EXPERIMENT

### 4.1 Model Setup

We compare two advanced pre-trained visual Transformer models as frozen visual encoders: (1) ViT-L/14 from CLIP [37], and (2) ViT-G/14 from EVA-CLIP [15]. Regarding the selection of frozen audio encoders, we investigate the audio encoders in ImageBind [19] and LanguageBind [59]. For frozen language models, we consider two different architectures of large language models: (1) FlanT5 [11], which is a large language model based on the encoder-decoder architecture, fine-tuned from T5 [38], and (2) Vicuna [10], which is a large language model based on a pure decoder architecture, fine-tuned from LLaMA instructions. We set the number of layers for the `Video Resampler` and `Audio Resampler` to 6, each containing 64 learnable queries with a dimension of 768. The number of layers for the `Audiovisual Resampler` is also set to 6, with 64 learnable queries with a dimension of 1,536. It is worth noting that during cross-modal pre-training and instruction fine-tuning stages, only the parameters of 3 perceivers and the LoRA module of the LLM are trainable. Table 2 summarizes the various model variants of PAVC-Reasoner along with their corresponding parameter counts. All experiments of PAVC-Reasoner are conducted on four Nvidia A100 (80GB) GPUs. Consistent pretraining hyperparameters were employed across all model variants. For more details on data preprocessing and training, we sincerely refer readers to Sup. B.

## 4.2 Automatic Evaluation with GPT-4

To comprehensively evaluate the quality of explanations generated by the model, we employed GPT-4 [1] for automatic assessment. Specifically, drawing inspiration from [10, 29], we input the model-generated explanations for a particular sample in specific testing scenarios, along with the gold standard annotations for the sample's explanation and a textual description of the visual context of the sample (encompassing detailed video descriptions, fine-grained object categories, and relevant physical attributes), into GPT-4. This prompts GPT-4 to score the model-generated explanations in terms of *accuracy*, *helpfulness*, *relevance*, and *granularity*. Scores range from 1 to 10, with higher scores indicating superior performance. We recorded the results of GPT-4's responses to the same explanation generated five times. To ensure the reliability of the scores, we also invited five human annotators to individually rate the explanations generated by the model under different testing scenarios (*e.g.* w/ audio and w/o audio) on a scale of 1-10 for extra assessment.

## 4.3 Baseline Methods

We considered three different types of baseline methods. Firstly, the simple late fusion (Q+I+V+A [51]) concatenates the embeddings of images, audio, video, and questions obtained from each single-modal encoder to form question-object embeddings. Subsequently, these question-object embeddings are concatenated and fed into an MLP to generate binary classification outputs. The second category comprises advanced multimodal foundation large Models such as AudioCLIP [20] and MERLOT Reserve [53]. It's worth noting that the first two categories of methods do not provide explanations. Lastly, we explored MLLMs as robust baselines, given their perceptual and cognitive abilities demonstrated in multimodal tasks. Specifically, the following MLLMs were considered: Open-Flamingo [3], LLaVA [29], mPLUG-Owl [47], VideoChat [25], Video-LLaMA [54], Video-ChatGPT [31], PandaGPT [41], and Video-LLaVA [28]. For most MLLM-style competitors, we employed an in-context learning multimodal prompt similar to that used in GPT-3 [6] and Flamingo [2] to achieve task adaptation, rather than fine-tuning them on instruction-following data. Due to limited space, we defer the implementation details of these methods to Sup. C.

## 4.4 Main Results

*4.4.1 Results on PACS-Material.* Table 3 presents the evaluation results of all comparison models on the PACS-Material and PACS-QA tasks. Firstly, we observe that the performance of all models on the PACS-Material, a binary classification task to identify object materials, surpasses that on the PACS-QA task by a considerable margin, with a performance gain (Δ) ranging between 10% and 20%. Considering the primary difference between these two tasks lies in the content of the questions, we can attribute this performance gap to the implicit queries about *physical properties of objects and multi-hop reasoning about physical commonsense* in the PACS-QA questions, which increases the difficulty to select correct answers. Secondly, PAVC-Reasoner outperforms all competitors in terms of accuracy, achieving new state-of-the-art on the two PACS tasks.

*4.4.2 Results on PACS-Reason.* Table 4 illustrates the fine-grained evaluation results of PAVC-Reasoner and baseline methods on

Table 4: *Performance comparisons among diverse multimodal large language model (MLLM)-based baselines on the PACS-Reason benchmark.* **Results are reported as the mean and standard deviation of five runs.**

| Baseline Model | Accuracy (%) | | Helpfulness (%) | | Relevance (%) | | Granularity (%) | | HumanEval (%) | |
| --- | --- | --- | --- | --- | --- | --- | --- | --- | --- | --- |
| | +audio | −audio | +audio | −audio | +audio | −audio | +audio | −audio | +audio | −audio |
| OpenFlamingo [3] | – | 65.6±1.3 | – | 71.6±1.2 | – | 75.2±1.3 | – | 72.4±1.9 | – | 63.7±1.2 |
| LLaVA [29] | – | 65.3±1.2 | – | 68.7±1.5 | – | 72.3±1.1 | – | 70.4±1.3 | – | 61.3±1.7 |
| mPLUG-Owl2 [48] | – | 67.3±1.5 | – | 72.3±1.8 | – | 77.4±1.6 | – | 74.8±1.6 | – | 64.5±1.0 |
| VideoChat [25] | – | 62.8±1.2 | – | 70.5±1.7 | – | 73.1±1.3 | – | 72.6±1.5 | – | 62.1±1.5 |
| Video-ChatGPT [31] | – | 64.5±1.6 | – | 71.2±0.6 | – | 74.3±1.2 | – | 73.2±1.1 | – | 62.9±1.3 |
| Video-LLaVA [28] | – | 66.6±1.6 | – | 71.9±0.7 | – | 75.3±1.1 | – | 73.5±0.8 | – | 63.8±1.8 |
| Video-LLaMA [54] | 63.4±1.5 | 62.9±0.3 | 75.5±1.9 | 70.2±0.2 | 77.4±0.9 | 75.1±0.8 | 76.2±0.7 | 73.5±1.2 | 65.6±1.2 | 62.6±1.1 |
| PandaGPT [41] | 64.1±1.4 | 63.5±1.9 | 76.7±2.4 | 70.9±1.4 | 78.3±1.8 | 75.7±1.8 | 76.9±1.6 | 74.6±1.9 | 66.2±1.1 | 63.3±1.4 |
| Video-LLaMA-2 [54] | 64.0±0.9 | 63.8±1.0 | 76.9±1.3 | 70.9±1.4 | 79.7±1.4 | 76.6±1.5 | 76.3±1.3 | 73.7±2.0 | 66.1±1.2 | 64.5±1.4 |
| PAVC-Reasoner-V2 | 76.2±1.5 | 73.8±1.6 | 77.7±1.0 | 72.6±1.1 | 80.4±0.8 | 78.3±0.5 | 76.7±2.1 | 74.2±0.6 | 68.8±1.3 | 64.6±0.5 |
| PAVC-Reasoner-V4 | 78.9±0.9 | 75.4±1.8 | 80.6±1.4 | 77.3±1.6 | 88.2±0.9 | 86.3±0.6 | 84.3±1.8 | 81.2±0.3 | 74.5±1.2 | 66.3±0.6 |

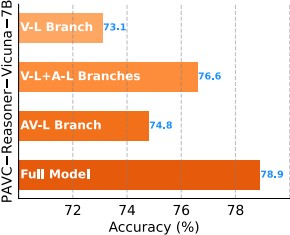 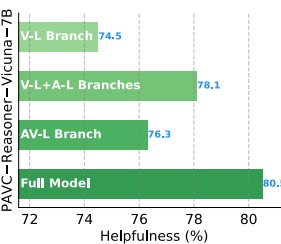 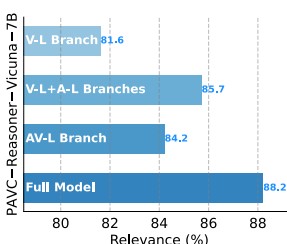 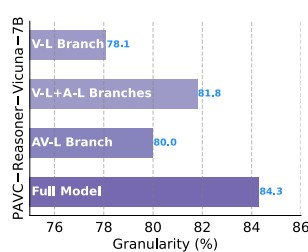

Figure 6: Ablation experiments on various perception branches (*x-language perceivers*) of PAVC-Reasoner.

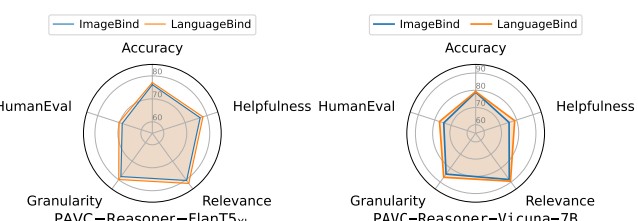

Figure 7: Ablations on different frozen audio encoders.

Table 5: Ablation studies of varying vision encoders and number of video sampling frames. The LLM is FlanT5$_{XL}$ [11].

| Branch | Visual Encoder | Frames | Accuracy (%) | HumanEval (%) |
| --- | --- | --- | --- | --- |
| V-L Branch | ViT-L/14 | 2 | 73.6 | 66.4 |
| | ViT-L/14 | 4 | 74.1 | 66.9 |
| | ViT-G/14 | 2 | 75.5 | 68.1 |
| | ViT-G/14 | 4 | 76.2 | 68.8 |

the PACS-Reason task. Overall, PAVC-Reasoner demonstrates outstanding performance across all evaluation metrics, indicating its effectiveness and superiority. Specifically, among our comparative methods, we include approaches similar to GPT-3 [6], which adapt MLLMs to the PACS task using in-context learning, as well as methods that fine-tune MLLMs using LoRA [22] on the same instruction data as PAVC-Reasoner. It is evident that our PAVC-Reasoner outperforms the competing MLLMs in terms of accuracy and the details of model-generated explanations. This can be attributed to i) the unique design of the *audiovisual-language perceiver* for capturing joint audiovisual information, which may be useful for identifying and understanding the physical properties of objects; ii) the multi-grained cross-modal pre-training significantly reduces the

gap between the visual/audio modalities and language modality. Similar trends can be found in Sup. D Table 1.

## 4.5 Ablation Study

*4.5.1 Effect of Three Perceivers.* To elucidate the contribution of each *perceiver* in PAVC-Reasoner, we conduct ablation experiments on the PACS-Reason task. As shown in Fig. 6, we start with a separate visual-language (V-L) perceiver. Given that images convey basic semantics such as object categories and shapes, the V-L perceiver empowers the LLM with basic visual content perception and comprehension capabilities. Subsequently, we find that the addition of an audio-language perceiver (A-L) improves the model's reasoning performance (acc. ↑ ∼3%). This may be due to the fact the introduction of audio data enables LLM to utilize auditory information to assist in recognizing objects in an image that are difficult to distinguish visually. Using only the audiovisual-language (AV-L) perceiver yields a modest performance improvement. Finally, the insertion of an AV-L perceiver further enhances the PAVC-Reasoner's understanding of joint audiovisual input signals, allowing it to more accurately discern physical properties between a pair of objects.

*4.5.2 Impact of Audio Encoders.* Fig. 7 compares the quality of predictions generated by model variants employing different frozen audio encoders. Specifically, we ablate ImageBind [19] (ViT-L/16, 307M) and LanguageBind [59] (ViT-L/14, 303M). The former aligns the embedding of each modality to the image embedding, while the latter directly aligns all modalities to the language space. The results on PACS-Reason show that the models equipped with the two different audio encoders produce negligible differences in predictions, with LanguageBind performing slightly better. This observation is because the *audio-language perceiver* in PAVC-Reasoner already

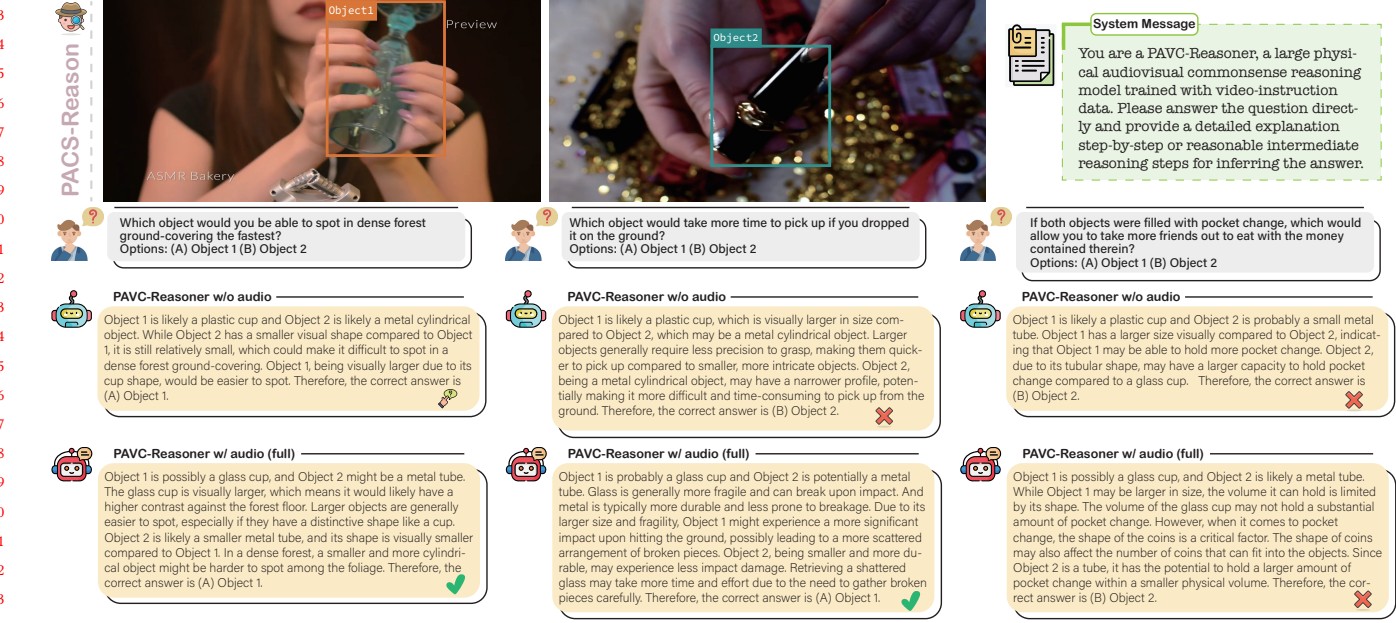

**Figure 8: Demonstration of the model-generated responses from the two variants of PAVC-Reasoner (i.e., PAVC-Reasoner-Vision 🤖 (w/o audio as input) and PAVC-Reasoner-V4 🤖 (full model, with audio as input)) on the PACS-Reason test set.**

aligns the audio with the language modality via cross-modal pre-training. We adopt LanugeBind as the default audio encoder.

*4.5.3 Impact of Vision Encoders.* Table 5 records the results of model variants equipped with different frozen visual encoders and sampling frame numbers on the PACS-Reason task. A more powerful visual encoder can extract finer details from images. It is evident that upgrading the visual encoder from ViT-L [37] to ViT-G[15] significantly improves both the accuracy of model inference (accuracy ↑ ~2%) and the quality of generated explanations. Additionally, we observe that decent inference performance can be achieved with fewer frames (*e.g.* 2-4). This is because the focus of PACS-Reason's questions is more on examining the material composition and physical properties of objects rather than emphasizing *video action recognition*. The former only requires a few key frames for judgment, while the latter demands more temporal information.

## 4.6 Case Study

Fig. 8 illustrates the physical commonsense reasoning capability of the PAVC-Reasoner variants in both non-audio and audible video scenarios, where PAVC-Reasoner-Vision 🤖 is equipped solely with a *visual-language perceiver*. PAVC-Reasoner-V4 🤖 includes the complete set of three *perceivers*. Both model variants use ViT-G/14 [15] as the visual encoder and Vicuna-7B [10] as the LLM.

Observing Fig. 8, the leftmost question only involves visual clues, where PAVC-Reasoner-Vision is capable of predicting the correct answer but generating erroneous explanations, such as incorrectly identifying Object1 as a plastic cup instead of a glass one (GT). This misidentification of object material can lead to incorrect predictions when answering questions involving auditory clues. For the middle question in Fig. 8, PAVC-Reasoner-Vision fails to be aware that glass objects can break and take longer to pick up from the ground.

Additionally, both models struggle to answer the rightmost question, which subtly inquires about the size and shape of two objects. While PAVC-Reasoner-Vision produces the correct rationale, it deduces the wrong answer. Conversely, PAVC-Reasoner-V4 generates incorrect explanations. We demonstrate more prediction examples of PAVC-Reasoner on PACS-Reason and the more challenging PACS-Reason+ in Sup. E Figs. 1-6. From these visualizations, we affirm three compelling features of PAVC-Reasoner: (*i*) the ability to decompose complex questions, discerning the requisite physical commonsense knowledge necessary to answer them; (*ii*) demonstrating a common understanding of the physical attributes and affordances of objects in videos; and (*iii*) performing complex commonsense reasoning based on joint audiovisual information–the ability to generate explanations for different types of questions.

## 5 CONCLUSION

We propose two augmented benchmarks, PACS-Reason and PACS-Reason+, for interpretable physical audio-visual reasoning. These benchmarks provide detailed explanations as rationales for each sample, enabling the assessment of the physical commonsense reasoning capability of multimodal models from multiple dimensions. Additionally, we introduce PAVC-Reasoner, a multimodal foundation model for reasoning about physical commonsense attributes. PAVC-Reasoner considers the collaboration and specificity of modalities, using three distinct perceivers to bridge vision/audio and language modalities via multi-grained cross-modal pretraining and instruction finetuning. It utilizes an LLM as a cognitive engine to comprehend multimodal inputs and generate convincing intermediate reasoning paths. Extensive experiments demonstrate the outstanding performance and effectiveness of PAVC-Reasoner.

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
