# OpenReview forum: "Toward Explainable Physical Audiovisual Commonsense Reasoning"
_acmmm.org/ACMMM/2024/Conference — MM2024 Oral_

### Official Review · Reviewer_p3ep · 2024-05-24

**Rating:** 5
**Confidence:** 3

**Summary:**

This paper explores explainable physical audiovisual commonsense reasoning tasks using large language models (LLMs). It introduces two new benchmarks, PACS-Reason and PACS-Reason+, which augment the PACS dataset by appending detailed reasoning explanations to each data point. Additionally, it proposes PAVC-Reasoner, a multi-modal foundational model fine-tuned with instructions for audio-visual commonsense reasoning. Finally, the extensive experiments prove the effectiveness of the proposed method.

**Strengths:**

1. The author introduces the explainable physical commonsense reasoning benchmarks, EPACS, based on PACS, utilizing foundational visual tools and GPT-4. These benchmarks provide meticulously crafted detailed explanations for answer choices, enabling a comprehensive evaluation of the physical commonsense reasoning capabilities of multimodal models.

2. The author proposes PAVC-Reasoner, which considers modality synergy and specificity, leveraging a large language model (LLM) as the core for processing multimodal signals and generating natural language rationales. Additionally, GPT-4 and human evaluation are used to assess reasoning accuracy from four aspects: accuracy, helpfulness, relevance, and granularity.

3. Extensive experiments, including comparisons against state-of-the-art (SOTA) models, ablation studies, and case studies, conclusively demonstrate the performance of the proposed method.

**Limitations:**

There are a few questions we hope the author can explain:
1. Section 3.1.3 Audiovisual-Language Perceiver. The author mentions that the video and audio representations,
$V$ and $A$, are processed by the unimodal encoders but does not explain how $V$ and $A$ are combined into a joint feature. Are they directly flattened and joined as one, or is there another special method used?

2. Whether the author considers reporting the performance of humans on EPACS. Table 4 lacks some comparison against humans for a better understanding of task hardness.

**Suitability:**

3

---

### Official Review · Reviewer_QyCM · 2024-05-25

**Rating:** 5
**Confidence:** 3

**Summary:**

In this paper, the authors focused on explainable physical audiovisual commonsense reasoning. They introduced two new benchmarks and a MLLM designed to reason about physical commonsense attributes. The proposed MLLM aligns different modalities with the language modality and utilizes a LLM as a cognitive engine to process multimodal inputs and generate outputs. Extensive experiments demonstrate the effectiveness of the proposed method.

**Strengths:**

There are three strong points:
1.	The authors introduced two benchmarks for explainable physical commonsense reasoning, which will significantly contribute to the development of Explainable Physical Audiovisual Commonsense (EPACS) systems.
2.	The designed MLLM leverages three perceivers to bridge multimodal inputs to the LLM, which serves as the core for processing inputs and generating natural language rationales.
3.	Experiments prove the effectiveness of the proposed method.

**Limitations:**

There are also some weak points:
1.	The perceiver design used in the MLLM is commonly employed in the MLLM community, such as Flamingo, thus the contribution from the model design perspective may be relatively limited.
2.	By integrating multimodal inputs, including audio and video, into the LLM, some crucial modality-specific characteristics may be lost. For example, certain audio features cannot be accurately described in natural language, potentially impacting performance.
3.	It would enhance the paper's impact if the constructed benchmarks were released to benefit the research community.

**Suitability:**

3

---

### Official Review · Reviewer_4zzp · 2024-05-27

**Rating:** 4
**Confidence:** 4

**Summary:**

The research introduces two datasets by extending the PACS QA dataset, PACS-Reason and PACS-Reason+, designed for evaluating explainable physical commonsense reasoning by providing detailed rationales for each answer. It also presents the PAVC-Reasoner, a multimodal model followed the Flamingo framework that integrates visual, audio, and audiovisual inputs through three distinct perceivers and leverages a large language model for generating natural language explanations. The PAVC-Reasoner undergoes cross-modal pretraining and instruction fine-tuning to improve its understanding and reasoning. Extensive experiments demonstrate its performance and effectiveness in reasoning about physical attributes and material compositions, marking a further step towards the task.

**Strengths:**

PACS-Reason and PACS-Reason+ provide detailed explanations (rationales) for answer choices, allowing for a comprehensive evaluation of the model's reasoning capabilities. The PACS-Reason+ extend the data scale by using the GPT4.


The PAVC-Reasoner model integrates visual, audio, and audiovisual information through three perceivers, which extended the Flamingo. This multimodal fusion is essential for tasks requiring comprehensive understanding and differentiation between similar objects.

**Limitations:**

The effectiveness of the evaluation benchmark relies heavily on the quality and variety of the data. The bias or limitation in the dataset could affect the model evaluation results, so the bias and rational quality analysis will helpful.

The prompting instructions should be listed or introduced, and its better to analyze the data quality and biases.

Suggest to complete relevant work with video commonsense benchmarks as (e.g., AGQA 2.0: An Updated Benchmark for Compositional Spatio-Temporal Reasoning, STAR: A Benchmark for Situated Reasoning in Real-World Videos)

The proposed method is very close to the vision language learning in the Flamingo framework with resamplers. Could you introduce the differences?

**Suitability:**

3

---

### Meta-Review · Area_Chair_L4jp · 2024-07-03

**Recommendation:** Accept (Oral)
**Confidence:** 5

**Metareview:**

The reviews of this submission are quite positive and reviewers are convinced by the authors' response. All the major concerns have been addressed. The final recommendation is accept.